# Electromagnetic scattering laws in Weyl systems

Ming Zhou[1], Lei Ying[1], Ling Lu[2], Lei Shi[3], Jian Zi[3] & Zongfu Yu[1]

Wavelength determines the length scale of the cross section when electromagnetic waves are scattered by an electrically small object. The cross section diverges for resonant scattering, and diminishes for non-resonant scattering, when wavelength approaches infinity. This scattering law explains the colour of the sky as well as the strength of a mobile phone signal. We show that such wavelength scaling comes from the conical dispersion of free space at zero frequency. Emerging Weyl systems, offering similar dispersion at non-zero frequencies, lead to new laws of electromagnetic scattering that allow cross sections to be decoupled from the wavelength limit. Diverging and diminishing cross sections can be realized at any target wavelength in a Weyl system, providing the ability to tailor the strength of wave–matter interactions for radiofrequency and optical applications.

[1] Department of Electrical and Computer Engineering, University of Wisconsin, Madison, Madison, 53705, USA. [2] Institute of Physics, Chinese Academy of Sciences and Beijing National Laboratory for Condensed Matter Physics, Beijing, 100190, China. [3] Department of Physics, Fudan University, Shanghai, 200433, China. Correspondence and requests for materials should be addressed to Z.Y. (email: zyu54@wisc.edu)

Electromagnetic scattering is a fundamental process that occurs when waves in a continuum interact with an electrically small scatterer. Scattering is weak under non-resonant conditions; an example is Rayleigh scattering, which is responsible for the colours of the sky. Conversely, scattering becomes much stronger with resonant scatterers, which have an internal structure supporting localized standing waves, such as antennas, optical nanoresonators, and quantum dots. Resonant scatterers have wide application because the resonance allows physically small scatterers to capture wave energy from a large area. As such, large electromagnetic cross sections, $\sigma$, are always desirable: a larger $\sigma$ value means, for example, stronger mobile phone signals[1] and higher absorption rates for solar cells[2].

The maximum cross section of resonant scattering is bounded by the fundamental limit of electrodynamics. One might be tempted to enlarge the scatterer to increase the cross section, but this strategy only works for non-resonant scattering, or electrically large scatterers. In resonant scattering, physical size only affects spectral bandwidth, while the limit of cross section is determined by the resonant wavelength $\lambda$ as[3]:

$$\sigma_{\mathrm{max}} = \frac{D}{\pi}\lambda^2 \qquad (1)$$

The directivity $D$ describes the anisotropy of the scattering; $D = 1$ for isotropic scatterers. Equation 1 shows that an atom[4] can have a $\sigma_{\mathrm{max}}$ similar to that of an optical antenna[5], despite the sub-nanometre size of the atom. This also means that optical scatterers cannot attain resonant cross sections as large as those of radiofrequency (RF) antennas, due to the smaller wavelengths involved.

**Fig. 1** The length scale of cross section and its relation to dispersion. **a** In free space, the resonant cross section scales according to $\sigma \sim \lambda^2$ or $\sim 1/\omega^2$. Large cross sections always favour low frequencies. For example, the cross section of an optical transition in an atom is small (~$10^{-12}$ m$^2$) because of the associated short wavelength (~µm). The cross section of an RF antenna is much larger (~$10^{-4}$ m$^2$) due to a much longer wavelength (~cm). Diverging cross sections are obtained around the DC point, which happens to be the apex of a conical dispersion. The double lines indicate double degeneracy due to polarization. **b** By embedding the resonant scatterer in a medium where the dispersion of the continuum exhibits conical dispersions located away from the DC point, diverging cross sections can be realized at high frequencies. The cross section scales according to $\sim 1/\Delta\omega^2$, where $\Delta\omega$ is the relative detuning between the resonant frequency of the scatterer and the Weyl point. In both **a** and **b**, regions with stronger colour indicate larger resonant scattering cross section. **c** Schematic of extraordinarily large cross section created by an electrically small resonant scatterer (red dot) placed inside Weyl photonic crystal

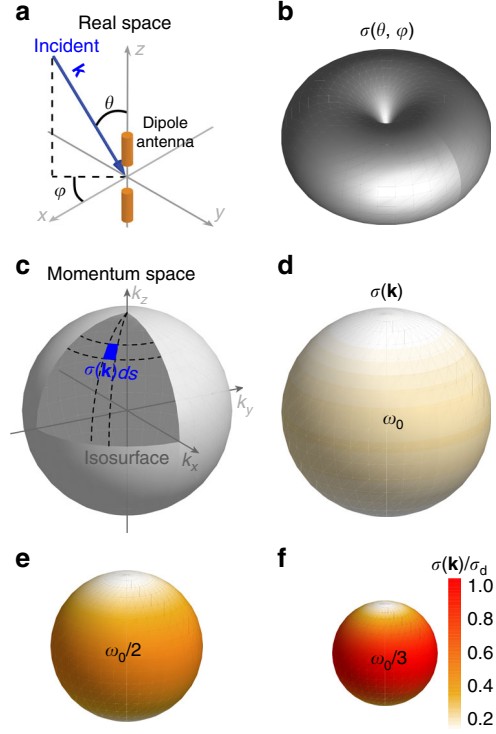

**Fig. 2** Scaling law of resonant cross sections in momentum space. **a** Schematic of a dipole antenna in free space. **b** Real-space representation of scattering cross section $\sigma(\theta,\varphi)$. **c** In the momentum space, the cross section is represented by $\sigma(\mathbf{k})$, with $\mathbf{k}$ located on the isosurface defined by the resonant frequency. **d** The cross section $\sigma(\mathbf{k})$ of the dipole antenna is represented by the colour intensity on the isosurface. **e** The cross section of a dipole antenna operating at a lower resonant frequency of $\omega_0/2$. The isosurface shrinks by half compared to **d**. The cross section increases as indicated by stronger colours. **f** Same as **e** but with an even smaller resonant frequency of $\omega_0/3$. **d**, **e** and **f** all use the same colour map so that the cross section can be directly compared. The cross section is normalized by $\sigma_d = 27\lambda_0^2/2\pi$

Overcoming the limit just described has far-reaching implications for RF and optoelectronic applications. Many efforts have been devoted to realizing this goal, including the use of enhanced directivity $D$[6–10], degenerate resonances[11], decreased dielectric constants[12] $\epsilon$ and materials with negative refractive index[13]. While these approaches exploit certain trade-offs to slightly increase the pre-factor in Eq. (1), the fundamental limit of $\lambda^2$ remains, which can be proven directly from Maxwell's equations without requiring specific scatterer details[5]. Until now, extremely large cross sections have only been obtained at long wavelengths near the DC frequency.

Here we show that the scattering laws in Weyl systems[14] allows the cross section to be decoupled from the wavelength limit. This opens a new path to realizing strong wave–matter interaction, providing potential benefits to RF and optoelectronic devices that rely on resonant scattering. Moreover, the scattering effect discovered here is equally applicable to acoustic or electronic waves[15–17].

## Results

### Length scale of the cross section

In free space, the dispersion directly leads to the wavelength limit of the cross section, which is shown by Eq. (1). Specifically, the DC point, which is located at the apex of the conical dispersion relation as shown in Fig. 1a, gives rise to the diverging cross sections at low frequencies. The

apex of the conic dispersion can be also realized at other spectral regimes, such as that shown in Fig. 1b. All the special scattering properties associated with the DC point can be reproduced, resulting in diverging cross sections and exceptionally strong light–matter interactions at high frequencies. Weyl points[14, 18–28], the three-dimensional (3D) analogy of Dirac points, have recently been shown to exhibit such conical dispersion relations. The scattering laws in Weyl systems allows the cross section to be decoupled from the wavelength limit. Extraordinarily large cross section can be realized even for very small scatterers inside a Weyl photonic crystal, which is schematically illustrated in Fig. 1c. Unlike the lensing effect that can concentrate incident waves to a fixed focus, the scatterer concentrates incident waves to itself no matter where it is placed inside the Weyl photonic crystal.

**Conservation law of resonant scattering.** To illustrate the underlying physics of the relation between the dispersion and cross section, we will first show a conservation law of resonant scattering. We start by considering the resonant cross section of a dipole antenna (Fig. 2a). Without losing generality, we only discuss the scattering cross section, assuming zero absorption. Similar conclusions can be drawn for the absorptive case, with the maximum absorption cross section $\frac{1}{4}$ that of the scattering cross section[29].

The dipole antenna is anisotropic due to its elongated shape, so the cross section $\sigma(\theta,\varphi)$ depends on the incident direction of the wave. At a normal direction, when $\theta = \pi/2$, it reaches its maximum value[5] of $\sigma_{\max} = 3\lambda^2/2\pi$. Along the axial direction, when $\theta = 0$, the cross section vanishes. While it is straightforward to calculate the cross section of a dipole antenna, it is not immediately apparent why the cross section follows the $\lambda^2$ rule and diverges around the DC frequency.

Figure 2b shows the real-space representation of $\sigma(\theta,\varphi)$ at the resonant frequency $\omega_0$. We can also represent $\sigma(\theta,\phi)$ in momentum space as $\sigma(\mathbf{k})$, with $\mathbf{k}$ located on the isosurface defined by $\omega = \omega_0$. As shown in Fig. 2c, the isosurface is a sphere with $|\mathbf{k}| = \omega_0/c$. To visualize the momentum-space representation, $\sigma(\mathbf{k})$ is indicated by the colour intensity on the isosurface in Fig. 2d. As we show in Supplementary Note 1, the resonant cross section satisfies the following conservation law:

$$\iint\limits_{s:\,\omega(\mathbf{k})=\omega_0} \sigma(\mathbf{k})\mathrm{d}s = 16\pi^2 \tag{2}$$

The integration is performed on the isosurface $\omega(\mathbf{k}) = \omega_0$. Our proof is based on quantum electrodynamics[30], so it applies to classical scatterers such as antennas, as well as to quantum scatterers such as electronic transitions that absorb and emit light. More importantly, the continuum, in which the scatterer is embedded, does not need to be free space; it can be anisotropic materials, or even photonic crystals[31], as long as a well-defined dispersion relation $\omega = \omega(\mathbf{k})$ exists.

Equation 2 dictates the scaling of $\sigma$ with respect to the resonant frequency $\omega_0$ of the scatterer. As $\omega_0$ decreases, the area of the isosurface shrinks. To conserve the value of the integration over a smaller isosurface, the cross section $\sigma(\mathbf{k})$ must increase accordingly. For example, Fig. 2e and f show the isosurfaces of dipole antennas with resonant frequencies at $\omega_0/2$ and $\omega_0/3$, respectively. The cross sections, indicated by colour intensity, must increase proportionally to maintain a constant integration over these smaller isosurfaces. This can also be seen by the average cross section in the momentum space:

$$\overline{\sigma} \equiv \frac{\iint \sigma(\mathbf{k})\mathrm{d}s}{\iint \mathrm{d}s} = \frac{16\pi^2}{S} \tag{3}$$

Here, $S \equiv \iint ds$ is the area of the isosurface. When approaching the apex of the conical dispersion relation, the isosurface diminishes, i.e., $S \to 0$. As a result, the cross section diverges around the DC point (Fig. 1a).

**Resonant scattering in Weyl photonic crystal.** Recent demonstrations of Weyl points in photonic crystals[18] show that the dispersion of a 3D continuum can exhibit conical dispersion at any designed frequency (Fig. 1b). The Hamiltonian for the continuum around the Weyl point is given by $\mathcal{H}(\mathbf{q}) = v_x q_x \sigma_x + v_y q_y \sigma_y + v_z q_z \sigma_z$, where $\sigma_{x,y,z}$ are Pauli matrices. The momentum $\mathbf{q}(q_x, q_y, q_z) = \mathbf{k} - \mathbf{k}_{\text{Weyl}}$ defines the distance to the Weyl point in the momentum space, and $\mathbf{q} = 0$ at the Weyl point. The linear dispersion described by this Hamiltonian produces an ellipsoidal isosurface that encloses the Weyl point. The isosurface shrinks to a point at the Weyl frequency $\omega_{\text{Weyl}}$. Scattering properties associated with the DC point are carried to high frequencies within Weyl medium, resulting in exceptionally strong resonant scattering. As illustrated in Fig. 1b, the average cross section scales as $\overline{\sigma} \sim \frac{1}{\left(\omega_0 - \omega_{\text{Weyl}}\right)^2}$ around the Weyl point. This allows high frequency resonant scatterers, such as atoms and quantum dots, to attain large cross sections, potentially at a macroscopic scale.

We now demonstrate a specific example of resonant scattering in a Weyl photonic crystal[22]. We consider a localized resonant scatterer in an infinitely large photonic crystal. The simulations are performed in two steps. First, we numerically calculate the eigenmodes of the resonant frequency in the Brillouin zone by using MIT photonic bands[32] (MPB). Next, we use each eigenmode as excitation and numerically calculate the scattering cross section by using the quantum scattering theory we developed recently[33], which is described in detail in Supplementary Note 2.

The structure of the Weyl photonic crystal consists of two gyroids, as shown in Fig. 3a. The magenta gyroid is defined by the equation $f(\mathbf{r}) > 1.1$, where $f(\mathbf{r}) = \sin(2\pi x/a)\cos(2\pi y/a) + \sin(2\pi y/a)\cos(2\pi z/a) + \sin(2\pi z/a)\cos(2\pi x/a)$ and $a$ is the lattice constant. The yellow gyroid is the spatial inversion of the magenta one. The two gyroids are filled with a material with a dielectric constant of $\epsilon = 13$. To obtain Weyl points, we add four air spheres to the gyroids to break the inversion symmetry (Fig. 3a). These spheres are related by an $S_4(z)$ transformation. The resulting band structure has four isolated Weyl points at the same frequency of $\omega_{\text{Weyl}} = 0.5645\,(2\pi c/a)$. Figure 3b illustrates conical dispersion in the $2k_z = -k_x - k_y$ plane.

The resonant scatterer is a quantum two-level system (TLS) embedded in the above photonic crystal. We numerically solve the scattering problem using quantum electrodynamics[33, 34]. The Hamiltonian is $= H_{\text{pc}} + H_{\text{TLS}} + H_{\text{I}}$. The first two terms, $H_{\text{pc}} = \sum_{\mathbf{q}} \hbar \omega_{\mathbf{q}} c_{\mathbf{q}}^{\dagger} c_{\mathbf{q}}$ and $H_{\text{TLS}} = \hbar \omega_0 b^{\dagger} b$, are the Hamiltonian of the photons and the TLS, respectively[35]. Here, $\hbar$ is the reduced Planck constant, $b^{\dagger}$ and $b$ are the raising and lowering operators for the quantum dot, respectively, and $c_{\mathbf{q}}^{\dagger}$ and $c_{\mathbf{q}}$ are the bosonic creation and annihilation operators of the photons, respectively. The Lamb shift[35] is incorporated into the resonant frequency $\omega_0$. The third term, $H_{\text{I}} = i\hbar \sum_{\mathbf{q}} g_{\mathbf{q}} (c_{\mathbf{q}}^{\dagger} b - c_{\mathbf{q}} b^{\dagger})$, is the interaction between the TLS and the radiation. The coupling coefficient is $g_{\mathbf{q}} = \mathbf{d} \cdot \hat{\mathbf{E}}_{\mathbf{q}} \sqrt{\omega_0 / 2\hbar \epsilon_0 L^3}$, where $\mathbf{d}$ is the dipole moment, $\epsilon_0$ is vacuum permittivity, $\hat{\mathbf{E}}_{\mathbf{q}}$ is the unit polarization vector of the photons and $L^3$ is the quantization volume.

Scattering inside photonic crystals is much more complex than that in free space. The continuum is highly dispersive, anisotropic and non-uniform, and thus the scattering cross section depends strongly on the location of the scatterer and orientation within the photonic crystal.

As an example, we consider a TLS with a transition frequency $\omega_0$ slightly below the Weyl frequency: $\omega_0 - \omega_{\text{Weyl}} = -0.0005\,(2\pi c/a)$. The calculated cross section $\sigma(\mathbf{q})$ (see Supplementary Note 2 for derivation) is plotted on the isosurface as shown by Fig. 3e, f. As expected, it strongly depends on the incident wavevector $\mathbf{q}$. In addition, $\sigma(\mathbf{q})$ varies greatly at different locations, as shown by comparing Fig. 3e and Fig. 3f. Despite all these differences, when integrated over the isosurface, $\iint \sigma(\mathbf{q}) ds$ always results in the same constant: $16\pi^2$. We perform the integration for a TLS at 20 different locations, all with the same constant, as shown in Fig. 3d.

As the transition frequency of the TLS approaches the Weyl point, i.e., $\omega_0 \to \omega_{\text{Weyl}}$, the isosurface shrinks in size, as illustrated by the insets of Fig. 4a. The conservation law leads to an increasing $\sigma(\mathbf{q})$, as shown by stronger colours. Near the Weyl frequency (black dashed line), the average cross sections $\overline{\sigma}$ is enhanced by three orders of magnitude compared to that in free space, eventually diverging at the Weyl point (Fig. 4a). The analytical prediction from Eq. 2 and the area of the isosurface agree very well with predictions from numerical simulation (circles in Fig. 4a).

**Frequency dependence of resonant scattering.** A Weyl point greatly enhances the cross section at the resonant frequency. However, it comes at the price of suppressed cross section away from the resonant frequency. Next, we discuss the spectral feature of the average cross section $\overline{\sigma}(\omega)$ for a given TLS. The spectral dependence is shown in Supplementary Note 3 as

$$\overline{\sigma}(\omega) \sim \frac{\left(\omega_0 - \omega_{\text{Weyl}}\right)^2 \wp}{\left(\omega_0 - \omega\right)^2 + \wp^2 \frac{\left(\omega_0 - \omega_{\text{Weyl}}\right)^4}{4}} \qquad (4)$$

Here, $\wp$ is a constant that depends on the local electric field at the position of the TLS, but does not vary significantly with frequency. In order to derive Eq. 4, we use the fact that the spontaneous decay rate is proportional to $\left(\omega_0 - \omega_{\text{Weyl}}\right)^2$ (Supplementary Eq. 32). At the resonance when $\omega_0 - \omega = 0$, the average cross section scales as $1/\left(\omega_0 - \omega_{\text{Weyl}}\right)^2$, which increases as the resonant frequency moves closer to the Weyl point. However, away from the resonance when $|\omega_0 - \omega| \gg |\omega_0 - \omega_{\text{Weyl}}|$, Eq. 4 reduces to $\overline{\sigma}(\omega_0) \sim \left(\omega_0 - \omega_{\text{Weyl}}\right)^2 / \left(\omega_0 - \omega\right)^2$, which shows that being close to the Weyl point suppresses the cross section. In Fig. 4b, we calculate the spectra for three different TLSs with their transition frequencies approaching the Weyl point (black dashed line). While the peak value of the cross section grows, the full width at half maximum of the spectrum decreases. The spectral integration of the cross section remains around a constant (Supplementary Note 3).

**Rayleigh or non-resonant scattering.** The non-resonant scattering of electrically small objects in free space follows the Rayleigh scattering law. In great contrast to resonant scattering, the cross section of Rayleigh scattering scales as $\sigma \sim \omega^4$, and diminishes at the DC frequency, as shown in Fig. 5a. This property is also closely related to the isosurface and can be carried to high frequency at the Weyl point (see proof in Supplementary Note 5). While the resonant cross section diverges, the non-resonant cross section diminishes at Weyl points, as illustrated in Fig. 5b.

Using perturbation theory and the first-order Born approximation[36], the Rayleigh scattering cross section can be shown as (details in Supplementary Note 5):

$$\sigma(\omega, \mathbf{k}_{\text{inc}}) = \omega^2 \iint_{S:\omega(\mathbf{k}_s)=\omega} d^2 \mathbf{k}_s \left| \mathbf{u}_{\mathbf{k}_s}(\mathbf{r}_0) V \mathbf{u}_{\mathbf{k}_{\text{inc}}}(\mathbf{r}_0) \right|^2 \propto \omega^2 S \qquad (5)$$

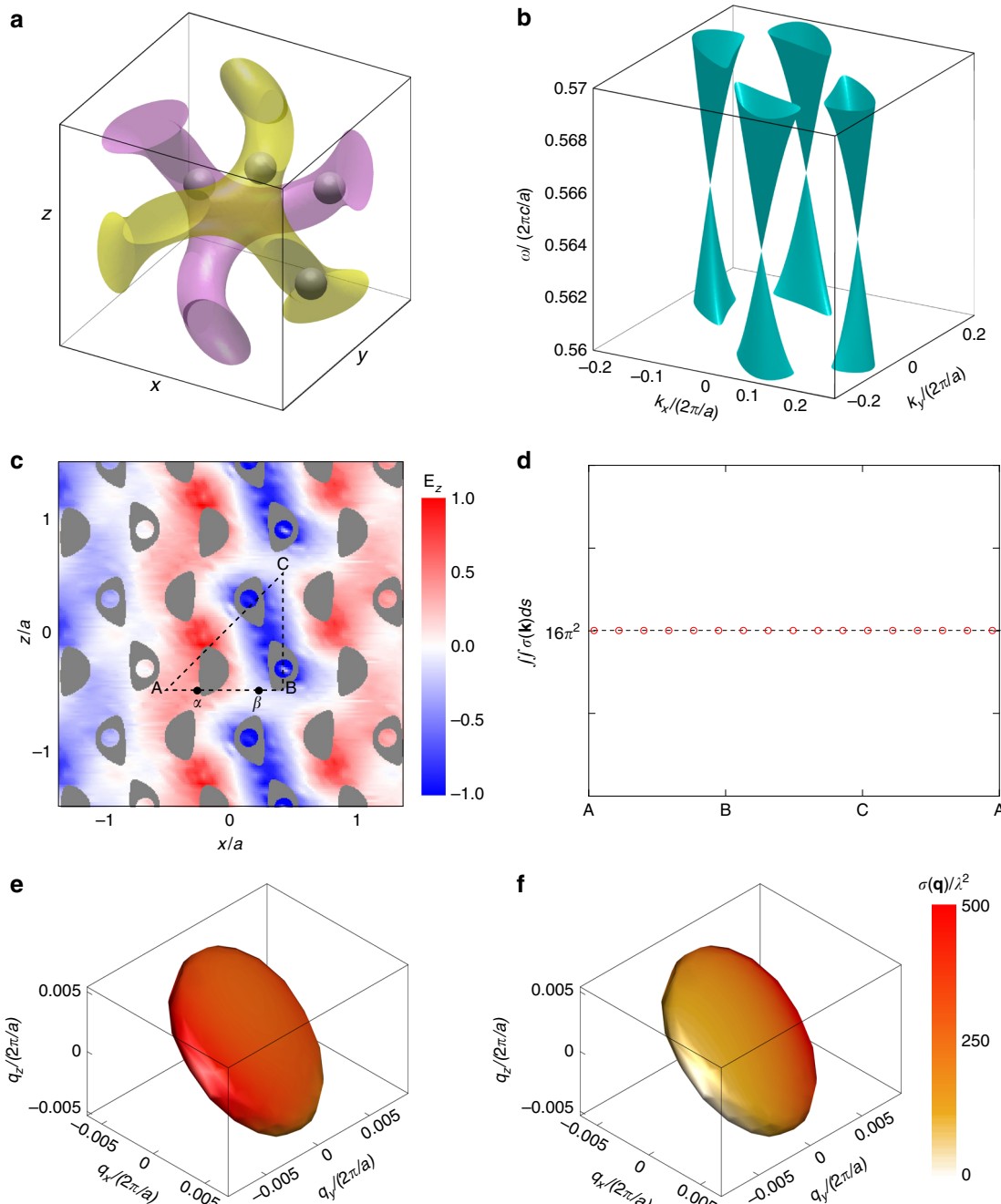

**Fig. 3** Simulation of resonant scattering in Weyl point photonic crystals. **a** The unit cell of the photonic crystal that supports Weyl points. Four air spheres with a radius of 0.07$a$, where $a$ is the lattice constant, are added to the gyroids to break the inversion symmetry. The positions of the air spheres are given by $\left(\frac{1}{4}, -\frac{1}{8}, \frac{1}{2}\right)a$, $\left(\frac{1}{4}, \frac{1}{8}, 0\right)a$, $\left(\frac{5}{8}, 0, \frac{1}{4}\right)a$ and $\left(\frac{3}{8}, \frac{1}{2}, \frac{1}{4}\right)a$. **b** Band structure of the photonic crystal close to the Weyl points. The band structure is plotted on the plane $2k_z = -k_x - k_y$. Four Weyl points are created with conical dispersion relation. **c** The normalized electric-field distribution of one eigenmode of the photonic crystal on the $x$–$z$ plane. We plot the $z$-component of the electric field as an example. **d** The resonant cross sections of the two-level system (TLS) for different locations. The integration in the momentum space always leads to the same constant. The dipole moment of the TLS is in the $x$ direction. Positions A-B-C-A are also labelled in **c**. **e**, **f** Examples of $\sigma(\mathbf{q})$ at two different positions $\alpha$ (**e**) and $\beta$ (**f**) are plotted on the isosurfaces as colour intensity. The positions $\alpha$ and $\beta$ are labelled in **c**. Both **e** and **f** use the same colour map so that the cross sections can be directly compared

Here, $\mathbf{u_k}(\mathbf{r})$ is the eigenmode of the Weyl photonic crystal associated with wavevector $\mathbf{k}$. $\mathbf{k_s}$ and $\mathbf{k}_{inc}$ are the wavevectors of the scattered and incident eigenmodes, respectively. $V$ is a tensor for the scattering potential of the Rayleigh scatterer. The integral is proportional to the area of the isosurface $S$. At the DC point, the isosurface shrinks to a point with $S = 0$, and the cross section of Rayleigh scattering is zero. Similarly, around the Weyl point, the area of the isosurface $S \sim \Delta\omega^2 = (\omega - \omega_{Weyl})^2$. The cross section scales as $\sigma \sim \omega^2 \Delta\omega^2$, and is zero at the Weyl point (Fig. 5b).

To validate our theoretical prediction above, we numerically calculate the Rayleigh scattering cross section of a small dielectric sphere embedded in the same Weyl photonic crystal. We use MPB[32] to obtain the eigenmode of the Weyl photonic crystal, then numerically calculate the Rayleigh scattering cross section using the normal-mode expansion[37–39] (see more details in

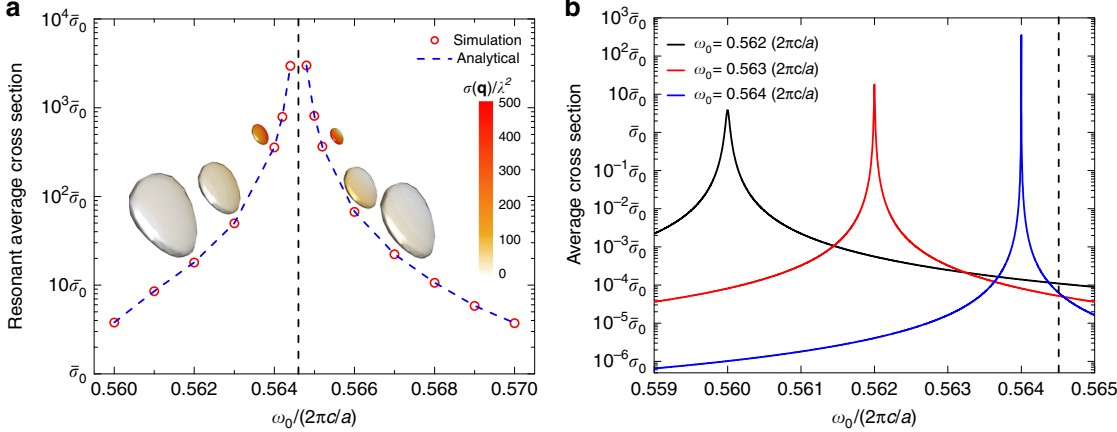

**Fig. 4** Resonant scattering cross section in Weyl photonic crystal. **a** Diverging resonant scattering cross section is realized around the Weyl frequency. Isosurfaces have an ellipsoidal shape (insets) with its colour indicating the value of the cross section; the shrinking isosurface leads to increasing cross sections around the Weyl frequency, which is indicated by the black dashed line. The results from quantum scattering simulation (red circles) agree well with the prediction based on the band structure (blue dashed line). The cross section is normalized by the average cross section in free space $\overline{\sigma}_0 = \lambda^2/\pi$. It scales as $\sigma \sim 1/\Delta\omega^2$. **b** Spectra of average scattering cross sections for TLSs with different transition frequencies. For simplicity, the spontaneous emission rates of the TLSs in vacuum are chosen to be $10^{-4}\omega_0$. We also assume the dipole moments and locations of the TLSs are the same. The black dashed line also indicates the Weyl frequency

Supplementary Note 6). The calculated Rayleigh scattering cross sections around the Weyl point are obtained by averaging 100 scatterers at random locations within one unit cell of the photonic crystal, and are plotted in Fig. 5c as red circles. They are normalized by $\sigma_R$, the scattering cross section of the same scatterer in free space. The blue dashed line indicates the scaling law $\sigma \sim \omega^2\Delta\omega^2$. The calculated Rayleigh scattering cross section agrees with the theoretical prediction well, with a vanishing cross section observed at the Weyl point. For high-frequency devices, such as integrated waveguides and laser cavities, Rayleigh scattering caused by interface roughness degrades performance and increases noise. The combination of suppressed Rayleigh scattering and enhanced resonant scattering could make Weyl media attractive for these optoelectronic applications.

## Discussion

Moreover, the conservation law of resonant scattering can also be extended to lower-dimensional space. In two-dimensional photonic crystals, diverging cross sections can be realized at Dirac points; an example is provided in Supplementary Note 4. To further generalize the findings in this paper, we may not necessarily need conical dispersion. Quadratic dispersion found around the band edges of photonic crystals also provides shrinking isosurfaces. However, this is less useful in practice because the zero group velocity at the band edge makes it difficult to obtain propagating waves[40] in the presence of disorders. In addition, coupling into such media is difficult due to the large impedance mismatch.

As a final remark, the transport properties of electrons around Dirac and Weyl points has also been studied in the past few years[15–17]. Some of the observations are consistent with the physics of photon scattering shown in this paper. Here, we explicitly show the general conservation law of cross section and its connection to the dispersion relation. We expect that similar conclusions can be drawn for both electrons and phonons. It provides useful insight to understand general scattering physics beyond Dirac and Weyl systems.

In conclusion, large resonant cross sections are of great practical importance. They are only achievable with long resonant wavelengths when the frequency approaches the DC point. As shown in this work, the dispersion, rather than the wavelength, is

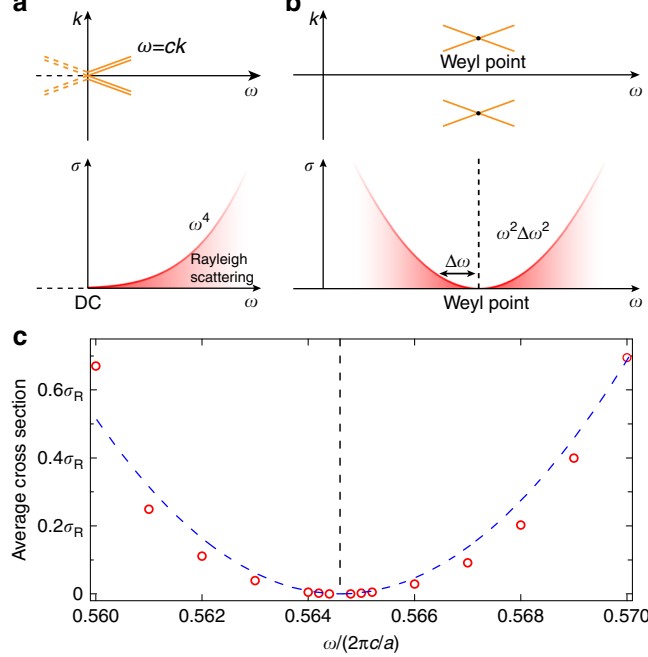

**Fig. 5** Rayleigh scattering in free space and Weyl systems. **a** In free space, the Rayleigh scattering cross section scales as $\sigma \sim \omega^4$ and vanishes at the DC frequency. **b** In Weyl systems, the Rayleigh scattering cross section scales as $\sigma \sim \omega^2\Delta\omega^2$ and vanishes at the Weyl frequency, which is indicated by the black dashed line. In both **a** and **b**, regions with stronger red colours indicate smaller Rayleigh scattering cross section. **c** Numerical calculation of Rayleigh scattering in a Weyl photonic crystal. The Rayleigh scatterer is a dielectric sphere with a radius of $0.01a$ and a dielectric constant of 2. The numerical results (red circles) are normalized by the Rayleigh scattering cross section in free space $\sigma_R$, and fit well the predicted $\sigma \sim \omega^2\Delta\omega^2$ scaling (blue dashed line). The black dashed line also indicates the Weyl frequency

responsible for the cross section. As a result, Weyl points, which can achieve similar conic dispersion as that around the DC point, lead to the diverging resonant cross section at any desired frequency. The exceptionally strong resonant scattering is also

accompanied by diminishing non-resonant scattering, which is also similar to that around the DC point. Since Weyl points can be realized at any frequency, we can effectively decouple the cross section and the wavelength. It opens up possibilities for tailoring wave–matter interaction with extraordinary flexibility, which also can be extended to acoustic and electronic wave scattering.

**Data availability**. The data that support the finding of this study are available from the corresponding author upon reasonable request.

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

## Acknowledgements

M.Z., L.Y. and Z.Y. acknowledge the finanical support of DARPA (YFA17 N66001-17-1-4049 and DETECT program). L.L. was supported by the National key R\&D Progam of China under Grant No. 2017YFA0303800, 2016YFA0302400 and supported by the National Natural Science fundation of China (NSFC) under Project No. 11721404. L.S. and J.Z. were supported by 973 Program (2015CB659400), China National Key Basic Research Program (2016YFA0301100) and NSFC (11404064).

## Author contributions

M.Z. developed the theoretical formalism, performed the analytic calculations and performed the numerical simulations. All authors contributed to the final versions of the manuscript. Z.Y. supervised the project.

## Additional information

**Competing interests:** The authors declare no competing financial interests.

