## [Peer Review File · Nature Communications]

Reviewers' comments:

Reviewer #1 (Remarks to the Author):

The authors study theoretically the scattering properties of an ideal two-level system with maximal scattering cross section placed in photonic crystal to control its optical properties via bandgap engineering. They found that in the vicinity of so-called Weyl points it is possible to decouple scattering cross section from the wavelength limit.

Starting from the abstract the authors emphasise that the scattering cross section diverges for resonant scattering system and exhibits non-Rayleigh behaviour. I don't agree with this statement, since in order to achieve resonant scattering condition in long wavelength limit, the effective size of the scatterer should be scaled accordingly. Thus, such conclusion will be correct for infinitely large scatterer, which eventually should produce diverging cross section, which appears to be an ill-defined problem. Some conditions were discussed in the literature where such limit can be achieved for meta material scatterers with negative permittivity and permeability - see, for example, PRA 80, 013808 (2009). If the authors would consider scattering efficiency instead of cross section, the limit should be finite. Second, an effective divergence of the scattering cross section in the vicinity of Weyl points can be attributed to increase of the local density of states. I wonder, how the authors performed their simulations of the periodic structure with a localised scatterer - it's not clear to me, did they consider just a single scatterer or they assumed periodically repeated scatterers in each unit cell? How the excitation of the scatterer was performed - by external excitation (like depicted in Fig.1) or by assuming that there is a mode, which is already somehow excited in the system?

Overall, I don't see strong evidence for the conclusions provided by the authors.

Reviewer #2 (Remarks to the Author):

Referee report on the manuscript 'Electromagnetic Scattering Laws in Weyl Systems' by Ming Zhou et al.

This is a theoretical work devoted to the study of light scattering in a photonic structure with Weyl point, where the dispersion $\omega(\mathbf{k})$ is linear in wave vector \mathbf{k} , $\omega - \omega_{\text{Weyl}} \propto \mathbf{\sigma} \cdot \mathbf{k}$, where $\mathbf{\sigma}$ is the vector of Pauli matrices.

It is demonstrated that for nonabsorbing scatterer the resonant scattering cross-section is enhanced at the Weyl point while the off-resonant (Rayleigh) scattering cross section is suppressed at the Weyl point. The results are corroborated by full-wave simulation as well as rigorous analytical derivation.

Overall, I find this work original and potentially relevant for Nature Communications. The results are solid. However, I am not quite convinced in the originality of this work with respect to previous studies in electronic systems and I think that the presentation can be improved. Therefore, I do not think that the manuscript is suitable for publication unless considerable revisions are made. My specific concerns are explained in detail in the attached pdf file.

Referee report on the manuscript “Electromagnetic Scattering Laws in Weyl Systems” by Ming Zhou et al.

This is a theoretical work devoted to the study of light scattering in a photonic structure with Weyl point, where the dispersion $\omega(\mathbf{k})$ is linear in wave vector \mathbf{k} , $\omega - \omega_{\text{Weyl}} \propto \boldsymbol{\sigma} \cdot \mathbf{k}$, where $\boldsymbol{\sigma}$ is the vector of Pauli matrices. It is demonstrated that for nonabsorbing scatterer the resonant scattering cross-section is enhanced at the Weyl point while the off-resonant (Rayleigh) scattering cross section is suppressed at the Weyl point. The results are corroborated by full-wave simulation as well as rigorous analytical derivation. Overall, I find this work original and potentially relevant for Nature Communications. The results are solid. However, I am not quite convinced in the originality of this work with respect to previous studies in electronic systems and I think that the presentation can be improved. Therefore, I do not think that the manuscript is suitable for publication unless considerable revisions are made. My specific concerns are explained below in detail.

- The Authors derive a nice expression S30 for the scattering cross section σ which very much reminds the Breit-Wigner formula for scattering in quantum physics,

$$\sigma \propto \frac{\Gamma}{(\omega_0 - \omega)^2 + \Gamma^2}, \quad \Gamma \propto (\omega - \omega_{\text{Weyl}})^2. \quad (1)$$

where Γ is the spontaneous emission rate, being proportional to the local density of (photonic) states (DOS). I believe it would be useful to present an equation of type (S30) directly in text and use it for explanation of the results. It is clearly seen from Eq. (1) that the resonant cross section at $\omega = \omega_0 = \omega_{\text{Weyl}}$ is proportional to $1/\Gamma$ and is increased when DOS tends to zero (i.e. at the Weyl point). On the other hand, the off-resonant cross section ($|\omega - \omega_0| \gg \Gamma$) is proportional to $\Gamma/(\omega_0 - \omega)^2$ and is suppressed at the Weyl point. I believe that discussion of such type of expression in text would make connection between off-resonant and resonant regimes more clear. It would be also beneficial to plot the spectra $\sigma(\omega)$ for different detuning of the scatterer with resonance from the Weyl point. Right now the classical Rayleigh and quantum resonant regimes are discussed in a rather different manner, while in fact they can be described universally, with the Rayleigh scattering corresponding to the Born approximation.

- The conservation laws of type Eq. (2) for the total cross section very much remind the optical theorem, well known in optics and in quantum mechanics (see e.g. Born and Wolf textbook or Landau and Lifshitz textbooks). I think that some reference is necessary here.
- Overall, the resonant enhancement of cross section in the Weyl or Dirac points is already well known for electronic systems, where the expressions similar to Eq. (1) have been obtained. For instance, Eq. (1) is identical to Eq. (31) in Nandkishore et al., “Rare region effects dominate weakly disordered three-dimensional Dirac points”, PRB **89**, 245110 (2014), where the resonances with vanishing linewidth near the Dirac point are explicitly discussed. This is essentially the same conclusion as in the considered manuscript. The suppressed conductivity due to enhanced scattering is also discussed in Tabert et al., “Optical and transport properties in three-dimensional Dirac and Weyl semimetals” PRB **93**, 085426 (2016), after Eq. (2). Moreover, there is a recent preprint Tobias Holder et al., “Electronic properties of disordered Weyl semimetals at charge neutrality”, arXiv:1704.05481v1, explicitly discussing the difference between the resonant and off-resonant scattering at the Weyl point.

This reviewer fully realizes that the study of photonic systems generally lags behind the condensed matter physics. The existence of earlier works on electrons does not necessarily preclude the generalization to photonic structures. Still, a convincing novelty justification is required to publish current results in such journal as Nature Communications and proper acknowledgement of previous works is necessary.

- The results in the manuscript appear to be quite general and applicable to any system with vanishing photonic DOS, not necessarily at the Weyl point. I wonder if the arguments of the authors about the scattering enhancement could be extended to other systems with low DOS, i.e. because of the photonic band gap or because of the resonant dispersion of the dielectric function.
- The argument about the scattering cross section enhancement at the Weyl point holds when the incident and scattered wave are measured *inside* the Weyl structure. What would happen in a real-life situation when a *finite-size* structure with Weyl dispersion is put in vacuum and the waves are sent and measured from outside? I have a

suspicion that the waves would not enter the structure exactly because of vanishing DOS. This would compensate the scattering enhancement and it is quite possible that no enhancement will actually take place in any experiment. Some numerical study of resonant scattering from a finite-size system would be highly beneficial.

- Misprint in the caption of Fig. 3: Simultion \rightarrow Simulation.

Referee: The authors study theoretically the scattering properties of an ideal two-level system with maximal scattering cross section placed in photonic crystal to control its optical properties via bandgap engineering. They found that in the vicinity of so-called Weyl points it is possible to decouple scattering cross section from the wavelength limit.

Response: We sincerely appreciate your time and efforts for reviewing our manuscript.

Referee: Starting from the abstract the authors emphasize that the scattering cross section diverges for resonant scattering system and exhibits non-Rayleigh behavior. I don't agree with this statement, since in order to achieve resonant scattering condition in long wavelength limit, the effective size of the scatterer should be scaled accordingly. Thus, such conclusion will be correct for infinitely large scatterer, which eventually should produce diverging cross section, which appears to be an ill-defined problem. Some conditions were discussed in the literature where such limit can be achieved for meta material scatterers with negative permittivity and permeability - see, for example, PRA 80, 013808 (2009). If the authors would consider scattering efficiency instead of cross section, the limit should be finite.

Response:

The scattering efficiency also diverges at DC point. The size of a dielectric resonator goes as λ [e.g. Petosa and Ittipiboon, "Dielectric Resonator Antennas: A Historical Review and the Current State of the Art," IEEE Antennas and Propagation Magazine, vol. 52, no. 5, pp. 91-116, (2010)] while the cross section goes as λ^2 .

Moreover, there are resonators with sizes independent of its resonant wavelength. For example, a LC circuit with lumped elements can have an extremely long resonant wavelength, but still maintains a small size ($\ll \lambda$). [e.g. Skrivervik et. Al., "PCS antenna design: the challenge of miniaturization," IEEE Antennas and Propagation Magazine, vol. 43, no. 4, pp. 12-27, (2001)].

Since the scattering efficiency depends on the specific implementation. We choose to discuss a more fundamental quantity – cross section. This is also preferred approach in many classical references [e.g. Chapter 2.16 of “Antenna Theory: Analysis and Design” by C. A. Balanis, 4th edition, Wiley (2016)].

Even when evaluating scattering efficiency, Weyl point leads to diverging scattering efficiency. Figure R1 shows the calculated scattering efficiency of a dielectric resonator. The scattering efficiency indeed goes to infinity at the Weyl point.

Fig. R1. Enhancement of scattering efficiency. We consider a dielectric sphere resonator with a scattering efficiency of 3 in vacuum.

Referee: Second, an effective divergence of the scattering cross section in the vicinity of Weyl points can be attributed to increase of the local density of states.

Response: This is incorrect. The density of states *decreases*, and approaches zero at Weyl points [e.g. Phys. Rev. B 83, 205101 (2011)] due to shrinking isosurface near Weyl points.

Also, the cross section is independent of *local* density of states. As we shown in Fig. 3c-3d in the manuscript, the integrated cross section is a constant regardless of the position in the crystal. Here for the first time, we showed that that cross section is determined by the isosurface in the momentum space. The isosurface is related to, but distinctly different from density of state. The density of states does not necessarily decrease when the isosurface decreases, and vice versa.

“

Figure 3. Simulation of the quantum scattering of a two-level system embedded in Weyl point photonic crystals. (left) The electric-field distribution of one eigenmode of the photonic crystal on the x - z plane. (right) The resonant cross sections of the TLS for different locations. The integration in momentum space always leads to the same constant. The dipole moment of the TLS is in the x direction. Positions A-B-C-A are also labeled in (left). Examples of $\sigma(\mathbf{q})$ at two different positions, α and β , are plotted on the isosurfaces as insets.

Referee: I wonder, how the authors performed their simulations of the periodic structure with a localized scatterer - it's not clear to me, did they consider just a single scatterer or they assumed periodically repeated scatterers in each unit cell? How the excitation of the scatterer was performed - by external excitation (like depicted in Fig.1) or by assuming that there is a mode, which is already somehow excited in the system?

Response: Thank you for pointing out these questions. We have added the above information into the revised manuscript. It reads as:

“

We now demonstrate a specific example of resonant scattering in a Weyl photonic crystal¹⁹. We consider a localized resonant scatterer in an infinitely large photonic crystal. The simulations are performed in two steps. First, we numerically calculate the eigenmodes of the resonant frequency in the Brillouin zone by using MIT Photonic Bands³² (MPB). Next, we use each eigenmode as excitation and numerically calculate the scattering cross section by using the quantum scattering theory we developed recently³³, which is described in detail in the Supplemental Information.

”

Referee: Overall, I don't see strong evidence for the conclusions provided by the authors.

Response: We thank the reviewer for many good questions. We hope that the conclusions of the papers are now better supported with more clarification and references.

Referee: This is a theoretical work devoted to the study of light scattering in a photonic structure with Weyl point, where the dispersion $\omega(\mathbf{k})$ is linear in wave vector \mathbf{k} , $\omega - \omega_{\text{Weyl}} \propto \boldsymbol{\sigma} \cdot \mathbf{k}$, where $\boldsymbol{\sigma}$ is the vector of Pauli matrices. It is demonstrated that for nonabsorbing scatterer the resonant scattering cross-section is enhanced at the Weyl point while the off-resonant (Rayleigh) scattering cross section is suppressed at the Weyl point. The results are corroborated by full-wave simulation as well as rigorous analytical derivation. Overall, I find this work original and potentially relevant for Nature Communications. The results are solid. However, I am not quite convinced in the originality of this work with respect to previous studies in electronic systems and I think that the presentation can be improved. Therefore, I do not think that the manuscript is suitable for publication unless considerable revisions are made. My specific concerns are explained below in detail.

Response: We are grateful to the reviewer for his/her quality time and effort spent on reviewing our manuscript. Here we below we will address these specific concerns, and further clarify the connection to electronic systems.

Referee: The Authors derive a nice expression S30 for the scattering cross section σ which very much reminds the Breit-Wigner formula for scattering in quantum physics,

$$\sigma \propto \frac{\Gamma}{(\omega_0 - \omega)^2 + \Gamma^2}, \quad \Gamma \propto (\omega - \omega_{\text{Weyl}}). \quad (1)$$

where Γ is the spontaneous emission rate, being proportional to the local density of (photonic) states (DOS). I believe it would be useful to present an equation of type (S30) directly in text and use it for explanation of the results. It is clearly seen from Eq. (1) that the resonant cross section at $\omega = \omega_0 = \omega_{\text{Weyl}}$ is proportional to $1/\Gamma$ and is increased when DOS tends to zero (i.e. at the Weyl point). On the other hand, the off-resonant cross section ($|\omega - \omega_0| \gg \Gamma$) is proportional to $\Gamma/(\omega_0 - \omega)^2$ and is suppressed at the Weyl point. I believe that discussion of such type of expression in text would make connection between off-resonant and resonant regimes more clear.

Response: We thank the reviewer for trying to understand the physics using the Breit-Wigner formula. Unfortunately, one typo in the equation quoted by the reviewer leads to misunderstanding of physics. The correct Breit-Wigner formula is

[see e.g. page 463, Section 6.7 of “Modern Quantum Physics” by J. J. Sakurai and J. Napolitano, 2nd edition, Addison-Wiley]

$$\sigma \propto 1/|\mathbf{k}|^2 \frac{\frac{\Gamma^2}{4}}{(\omega - \omega_0)^2 + \frac{\Gamma^2}{4}}, \quad (\text{R1})$$

The numerator is Γ^2 , instead of Γ . As a result, there is no connection between the maximum cross section σ and spontaneous emission rate Γ . The part of $\frac{\frac{\Gamma^2}{4}}{(\omega - \omega_0)^2 + \frac{\Gamma^2}{4}}$ only control the lineshape.

For example, considering two dipoles with the same resonant wavelength of λ , but different dipole moments of \mathbf{d} and $2\mathbf{d}$. Their spontaneous emission rates are Γ and 4Γ , respectively. However, their maximum resonant cross section is always the same, *i.e.* $3\lambda^2/2\pi$.

The physics of increased cross section cannot be reflected in the original Breit-Wigner formula because it uses simple free space to derive the prefactor $1/|\mathbf{k}|^2$, which is basically λ^2 . In a way, we derive the generalized Breit-Wigner formula.

Here, we have added a new paragraph after Eq. S30 to show the connection to Breit-Wigner formula.

Referee: It would be also beneficial to plot the spectra $\sigma(\omega)$ for different detuning of the scatterer with resonance from the Weyl point.

Response: This is a great point and we thank you for bringing it to us. We have plotted the spectral $\sigma(\omega)$ for different detuning of the scatterer and added a new panel to Fig. 4. where $\sigma(\omega)$ is averaged over all incident directions. The revised description of Fig.4 reads:

Figure 4. Resonant scattering cross section in Weyl photonic crystal. (a) Spectrum of average scattering cross section for TLSs with different transition frequencies. For simplicity, the spontaneous emission rates of the TLSs in vacuum are chosen to be $10^{-4}\omega_0$. (b) Diverging resonant scattering cross section is realized around the Weyl frequency. Isosurfaces have an ellipsoidal shape (insets) with its color indicating the value of the cross section; shrinking isosurface leads to increasing cross sections around the Weyl frequency. The results from quantum scattering simulation (red circles) agree well with the prediction based on the band structure (blue dashed line). The cross section is normalized by the average cross section in free space $\bar{\sigma}_0 = \lambda^2/\pi$. It scales as $\sigma \sim 1/\Delta\omega^2$.

We further calculate the spectrum of the average cross section for TLSs with different transition frequencies and plot the spectrum in Fig. 4a. As the TLS's transition frequency approaches the Weyl point, *i.e.*, $\omega_0 \rightarrow \omega_{Weyl}$, the isosurface shrinks in size, as illustrated by the insets of Fig. 4b. The conservation law leads to an increasing $\sigma(\mathbf{q})$, as shown by stronger colors. Consequently, the average cross section increases, as shown by higher resonant peaks in Fig. 4a. Near the Weyl frequency, the average cross sections $\bar{\sigma}$ is enhanced by three orders of magnitude compared to that in free space, eventually diverging at the Weyl point (Fig. 4b). The analytical prediction from Eq. 2 and the area of the isosurface agree very well with predictions from numerical simulation (circles in Fig. 4b).

”

Referee: Right now the classical Rayleigh and quantum resonant regimes are discussed in a rather different manner, while in fact they can be described universally, with the Rayleigh scattering corresponding to the Born approximation.

Response: Thank you for this great suggestion. We have added a new section in the Supplemental Information to prove that Rayleigh scattering is suppressed in Weyl systems by using the first-order Born approximation. It reads as:

“

4. Proof of suppressed Rayleigh (non-resonant) scattering in Weyl systems

The suppressed Rayleigh scattering in Weyl system can be directly proved using perturbation theory and Born approximation. In this section, we will describe the mathematical proof in detail.

Under the framework of first-order Born approximation⁸, the scattering amplitude $f(\mathbf{k}_s, \mathbf{k}_{inc})$ for a Rayleigh scatterer with a weak scattering potential $\mathbf{V}(\mathbf{r})$ is given by

$$f(\mathbf{k}_s, \mathbf{k}_{inc}) = \int d^3\mathbf{r}' \mathbf{u}_{\mathbf{k}_s}(\mathbf{r}') \mathbf{V}(\mathbf{r}') \mathbf{u}_{\mathbf{k}_{inc}}(\mathbf{r}') e^{i(\mathbf{k}_{inc} - \mathbf{k}_s) \cdot \mathbf{r}'} \quad (\text{S33})$$

Here $\mathbf{u}_{\mathbf{k}} e^{i\mathbf{k} \cdot \mathbf{r}}$ is the eigenmode associated with wave vector \mathbf{k} . \mathbf{k}_s and \mathbf{k}_{inc} are the wave vectors of the scattered and incident eigenmodes, respectively. For simplicity, we assume the scatterer is very small compared to the incident wavelength and can be considered as a point scatter located at \mathbf{r}_0 . The scattering potential then can be written as $\mathbf{V}(\mathbf{r}) = \omega \mathbf{V} \delta(\mathbf{r} - \mathbf{r}_0)$. Note here \mathbf{V} is a tensor containing the dielectric constant of the scatterer. Thus, the scattering amplitude becomes

$$f(\mathbf{k}_s, \mathbf{k}_{inc}) = \omega \mathbf{u}_{\mathbf{k}_s}(\mathbf{r}_0) \mathbf{V} \mathbf{u}_{\mathbf{k}_{inc}}(\mathbf{r}_0) e^{i(\mathbf{k}_{inc} - \mathbf{k}_s) \cdot \mathbf{r}_0} \quad (\text{S34})$$

The total scattering cross section then can be calculated as

$$\sigma(\omega, \mathbf{k}_{inc}) = \sum_{\mathbf{k}_s} |f(\mathbf{k}_s, \mathbf{k}_{inc})|^2 = \omega^2 \sum_{\mathbf{k}_s} |\mathbf{u}_{\mathbf{k}_s}(\mathbf{r}_0) \mathbf{V} \mathbf{u}_{\mathbf{k}_{inc}}(\mathbf{r}_0)|^2 \quad (\text{S35})$$

The summation over \mathbf{k}_s can be readily converted to a surface integral over the isosurface given by $\omega(\mathbf{k}_s) = \omega$, and Eq. Sx becomes

$$\sigma(\omega, \mathbf{k}_{inc}) = \omega^2 \iint_{S: \omega(\mathbf{k}) = \omega} d^2\mathbf{k}_s |\mathbf{u}_{\mathbf{k}_s}(\mathbf{r}_0) \mathbf{V} \mathbf{u}_{\mathbf{k}_{inc}}(\mathbf{r}_0)|^2 \propto \omega^2 S \quad (\text{S36})$$

where S is the area of the isosurface. In free space, the area of the isosurface S scales $S \sim \omega^2$. Consequently, the Rayleigh scattering cross section scales as $\sigma(\omega, \mathbf{k}_{inc}) \sim \omega^4$. In great contrast, the area of the isosurface S scales $S \sim \Delta\omega^2 = (\omega - \omega_{Weyl})^2$ and thus the Rayleigh scattering cross section scales as $\sigma(\omega, \mathbf{k}_{inc}) \sim \omega^2 (\omega - \omega_{Weyl})^2$ in Weyl systems.

”

We also revised the description in the manuscript. Now it reads as:

“

Using perturbation theory and the first-order Born approximation³⁶, the Rayleigh scattering cross section can be shown as (details in the Supplemental Information)

$$\sigma(\omega, \mathbf{k}_{inc}) = \omega^2 \iint_{S:\omega(\mathbf{k}_s)=\omega} d^2\mathbf{k}_s |\mathbf{u}_{\mathbf{k}_s}(\mathbf{r}_0) \mathbf{V} \mathbf{u}_{\mathbf{k}_{inc}}(\mathbf{r}_0)|^2 \propto \omega^2 S \quad (4)$$

Here $\mathbf{u}_{\mathbf{k}}(\mathbf{r})$ is the eigenmode of the Weyl photonic crystal associated with wavevector \mathbf{k} . \mathbf{k}_s and \mathbf{k}_{inc} are the wavevectors of the scattered and incident eigenmodes, respectively. \mathbf{V} is a tensor for the scattering potential of the Rayleigh scatterer. The integral is proportional to the area of the isosurface S . At the DC point, the isosurface shrinks to a point with $S = 0$, and the cross section of Rayleigh scattering is zero. Similarly, around the Weyl point, the area of isosurface $S \sim \Delta\omega^2 = (\omega - \omega_{Weyl})^2$. The cross section scales as $\sigma \sim \omega^2 \Delta\omega^2$, and is zero at the Weyl point (Fig. 5b).

”

Referee: The conservation laws of type Eq. (2) for the total cross section very much remind the optical theorem, well known in optics and in quantum mechanics (see e.g. Born and Wolf textbook or Landau and Lifshitz textbooks). I think that some reference is necessary here.

Response: Thank you for pointing out the missing reference. We have added the following reference into the revised manuscript.

“

30. Berestetskii, V. B., Pitaevskii, L. P. & Lifshitz, E. M. *Quantum Electrodynamics, Second Edition: Volume 4*. (Butterworth-Heinemann, 1982).

”

Referee: Overall, the resonant enhancement of cross section in the Weyl or Dirac points is already well known for electronic systems, where the expressions similar to Eq. (1) have been obtained. For instance, Eq. (1) is identical to Eq. (31) in Nandkishore et al., “Rare region effects dominate weakly disordered three-dimensional Dirac points”, PRB 89, 245110 (2014), where the resonances with vanishing linewidth near the Dirac point are explicitly discussed. This is essentially the same conclusion as in the considered manuscript. The suppressed conductivity due to enhanced scattering is also discussed in Tabert et al., “Optical and transport properties in three-dimensional Dirac and Weyl semimetals” PRB 93, 085426 (2016), after Eq. (2). Moreover, there is a recent preprint Tobias Holder et al., “Electronic properties of disordered Weyl semimetals at charge neutrality”, arXiv:1704.05481v1, explicitly discussing the difference between the resonant and off-resonant scattering at the Weyl point.

This reviewer fully realizes that the study of photonic systems generally lags behind the condensed matter physics. The existence of earlier works on electrons does not necessarily preclude the generalization to photonic structures. Still, a convincing novelty justification is required to publish current results in such journal as Nature Communications and proper acknowledgement of previous works is necessary.

Response:

We appreciate the reviewer’s perspective on the progress made in electronic and photonic systems. We also agree that “the existence of earlier works on electrons does not necessarily preclude the generalization to photonic structures.”

Now we try to articulate the novelty compared to existing literature in electronic systems. While it is recently known that the transport of electrons around Dirac and Weyl points exhibits unusual behaviors, here we for the first time derive the general conservation law of the cross section $\iint_{s: \omega(\mathbf{k})=\omega_0} \sigma(\mathbf{k}) ds = 16\pi^2$. It applies not only to Dirac or Weyl points, but also to any medium with well defined dispersion. It provides the most general connection between the dispersion relation and the cross section. We believe it will also help condensed matter physicists with new perspectives to understand the conductance of electrons in materials.

As a minor note, Eq. (1), being the same as that in PRB 89, 245110 (2014), is *not* our new result. We quoted it as a brief review of scattering physics.

We have added following references into the revised manuscript

“

26. Nandkishore, R., Huse, D. A. & Sondhi, S. L. Rare region effects dominate weakly disordered three-dimensional Dirac points. *Phys. Rev. B* **89**, 245110 (2014).

27. Tabert, C. J., Carbotte, J. P. & Nicol, E. J. Optical and transport properties in three-dimensional Dirac and Weyl semimetals. *Phys. Rev. B* **93**, 085426 (2016).

28. Holder, T., Huang, C.-W. & Ostrovsky, P. Electronic properties of disordered Weyl semimetals at charge neutrality. *ArXiv170405481 Cond-Mat* (2017).

”

We have also added a new paragraph into the revised manuscript. It reads as

“

As a final remark, the transport properties of electrons around Dirac and Weyl points has also been studied in past few years²⁶⁻²⁸. Some of the observations are consistent with the physics of photon scattering shown in this paper. Here, we explicitly show the general conservation law of cross section and its connection to the dispersion relation. We expect similar conclusion can be drawn for both electrons and phonons. It provides useful insight to understand general scattering physics beyond Dirac and Weyl systems.

”

Referee: The results in the manuscript appear to be quite general and applicable to any system with vanishing photonic DOS, not necessarily at the Weyl point. I wonder if the arguments of the authors about the scattering enhancement could be extended to other systems with low DOS, i.e. because of the photonic band gap or because of the resonant dispersion of the dielectric function.

Response: This is a great question. We have discussed the generalization of our findings in the manuscript. It reads as

“

To further generalize the findings in this paper, we may not necessarily need conical dispersion. Quadratic dispersion found around the band edges of photonic crystals, also provides shrinking isosurfaces. However, this is less useful in practice because the zero group velocity at the band edge makes it difficult to obtain propagating waves³⁹ in the presence of disorders. In addition, coupling into such media is difficult due to the large impedance mismatch.

”

Referee: The argument about the scattering cross section enhancement at the Weyl point holds when the incident and scattered wave are measured inside the Weyl structure. What would happen in a real-life situation when a finite-size structure with Weyl dispersion is put in vacuum and the waves are sent and measured from outside? I have a suspicion that the waves would not enter the structure exactly because of vanishing DOS. This would compensate the scattering enhancement and it is quite possible that no enhancement will actually take place in any experiment. Some numerical study of resonant scattering from a finite-size system would be highly beneficial.

Response: It has been experimentally shown that coupling into the Weyl photonic crystal is not an issue. In an experiment shown in Fig. R1, which is taken from a previous work of one of the co-authors, the transmission around the Weyl frequency (~11.3 GHz) is up to 10%. It can be further improved with anti-reflection layer. The only drawback of coupling into Weyl photonic crystal is that the incident light has to come in from a particular angle.

C Transmission data

Fig. R1 Transmittance of Weyl photonic crystal slab. Plot obtained from Lu et. Al. "Experimental observation of Weyl points" Science, Vol. 349, no. 6248, pp. 622-624 (2015).

Referee: Misprint in the caption of Fig. 3: Simultion \rightarrow Simulation.

Response: Thank you for pointing it out. We have corrected the typo.

Reviewers' comments:

Reviewer #1 (Remarks to the Author):

I'm pleased with the authors replies on all comments and suggestion mentioned by Referees and don't have any further ones. They have revised their manuscript accordingly. I can recommend it for publication now in the current form.

Reviewer #2 (Remarks to the Author):

Referee report on the manuscript "Electromagnetic Scattering Laws in Weyl Systems" by Ming Zhou et al (2nd round).

I do appreciate the efforts of the authors to improve the revised manuscript. I am inclining to recommend the publication.

I still have a feeling that some of my comments in the previous reports were not examined in detail. So I want first to clarify the remaining questions on the frequency dependence of the cross section and of the spontaneous decay rate.

The Authors write: "We thank the reviewer for trying to understand the physics using the Breit-Wigner formula. Unfortunately, one typo in the equation quoted by the reviewer leads to misunderstanding of physics" and then present the Breit-Wigner expression Eq. R1. I do not argue with the correctness of the textbook Eq. R1 but I still insist that the *integral* cross section is proportional to

$$\sigma \sim \Gamma / ((\omega - \omega_0)^2 + \Gamma^2/4),$$

with Γ (not Γ^2) in the numerator.

To illustrate this point I suggest more close examination of Eq. S31 in the manuscript. The spontaneous decay rate Γ is proportional to $\sum g^2$ (line after Eq S29), and, according to 1st line of Eq S31, is proportional to $(\omega - \omega_0)^2$ as well. Hence, the factor $(\omega - \omega_0)^2$ in the denominator of Eq. S31 cancels out. The integral cross section has the form

$$\sigma \sim \Gamma / [(\omega - \omega_0)^2 + \Gamma^2/4]$$

or

$$\sigma \sim (\omega_0 - \omega_W)^2 / [(\omega - \omega_0)^2 + (\omega_0 - \omega_W)^2/4].$$

This is exactly Eq (1) from my 1st report, a derived directly from the results of the Authors. And it is equivalent to Eq. 31 from Nandkishore et al, so it by no means original.

In this form it is clear that the off-resonant cross section decreases when approaching the Weyl point ($\sigma \sim \Gamma / (\omega - \omega_0)^2$), while the resonant one increases ($\sigma \sim 1/\Gamma$, since $\Gamma \rightarrow 0$ when $\omega_0 = \omega_W$).

The area under the cross section spectrum stays the same when ω_0 changes. Exactly at the Weyl point, $\omega_0 = \omega_W$, the cross section has the delta-function shape, $\delta \sim \sigma(\omega_0 - \omega_W)$. This is exactly the result of the manuscript under consideration.

In order to elucidate this point better I have plotted the cross section spectra for different detunings $\omega_0 - \omega_W$, see the attached pdf. It is clear from the calculation that (i) the width of the

peak decreases and (ii) the height increases at the Weyl point which reflects decrease of the spontaneous decay rate Γ . The area under the peak stays the same. I believe that the parameters in my calculation illustrate the points (i) slightly better than in Fig. 4a of the manuscript, where the change of the linewidth with the detuning is not resolved.

To summarize, I recommend the Authors to present and discuss carefully and explicitly frequency dependences of *both* spontaneous spectral linewidth and integral cross section (not only the cross section) as functions of $\omega_0 - \omega_W$ and $\omega_q - \omega_0$, both in the main text and in the Supplementary. No big changes are required in the structure of the manuscript. However, such discussion would be really helpful for future readers and would justify the concluding statement "Here, we explicitly show the general conservation law of cross section and its connection to the dispersion relation".

```
> restart;
> g:=(omega[0]-omega[W])^2;
```

$$g := (\omega_0 - \omega_W)^2 \quad (1)$$

```
> sigma:=1/(omega[0]-omega[W])^2*g^2/((omega[q]-omega[0])^2+g^2/4);
```

$$\sigma := \frac{(\omega_0 - \omega_W)^2}{(\omega_q - \omega_0)^2 + \frac{1}{4}(\omega_0 - \omega_W)^4} \quad (2)$$

```
> plots[logplot]([subs(omega[W]=1,omega[0]=0.9,sigma),subs(omega[W]=1,omega[0]=0.98,sigma),subs(omega[W]=1,omega[0]=0.999,sigma)],omega[q]=0.8..1.1,legend=[omega[0]=0.9,omega[0]=0.98,omega[0]=0.999],labels=[omega[q]-omega[0],'sigma'],axes=boxed,font=[Times,16],labelfont=[Times,16]);
```

Reviewer 1

Referee: I'm pleased with the authors replies on all comments and suggestion mentioned by Referees and don't have any further ones. They have revised their manuscript accordingly. I can recommend it for publication now in the current form.

Response: We sincerely appreciate your time and efforts for reviewing our manuscript.

Referee: I do appreciate the efforts of the authors to improve the revised manuscript.

I am inclining to recommend the publication.

Response: We thank the reviewer for his/her continued support.

Referee: I still have a feeling that some of my comments in the previous reports were not examined in detail. So I want first to clarify the remaining questions on the frequency dependence of the cross section and of the spontaneous decay rate.

The Authors write: “We thank the reviewer for trying to understand the physics using the Breit- Wigner formula. Unfortunately, one typo in the equation quoted by the reviewer leads to misunderstanding of physics” and then present the Breit-Wigner expression Eq. R1.

I do not argue with the correctness of the textbook Eq. R1 but I still insist that the *integral* cross section is proportional to

$$\sigma \sim \frac{\Gamma}{(\omega_q - \omega_0)^2 + \frac{\Gamma^2}{4}}$$

with Γ (not Γ^2) in the numerator.

To illustrate this point I suggest more close examination of Eq. S31 in the manuscript.

The spontaneous decay rate Γ is proportional to $\sum g^2$ (line after Eq S29), and, according to 1st line of Eq S31, is proportional to $(\omega_{Weyl} - \omega_0)^2$ as well.

Hence, the factor $(\omega_{Weyl} - \omega_0)^2$ in the denominator of Eq. S31 cancels out. The integral cross section has the form

$$\sigma \sim \frac{\Gamma}{(\omega_q - \omega_0)^2 + \frac{\Gamma^2}{4}}$$

or

$$\sigma \sim \frac{(\omega_0 - \omega_{Weyl})^2}{(\omega_q - \omega_0)^2 + \frac{(\omega_0 - \omega_{Weyl})^4}{4}}$$

This is exactly Eq (1) from my 1st report, a derived directly from the results of the Authors. And it is equivalent to Eq. 31 from Nandkishore et al, so it by no means original.

In this form it is clear that the off-resonant cross section decreases when approaching the Weyl point ($\sigma \sim \Gamma / (\omega_q - \omega_0)^2$), while the resonant one increases ($\sigma \sim 1/\Gamma$, since $\Gamma \rightarrow 0$ when $\omega_0 = \omega_{Weyl}$).

The area under the cross section spectrum stays the same when ω_0 changes.

Exactly at the Weyl point, $\omega_0 = \omega_{Weyl}$, the cross section has the delta-function shape, $\sigma \sim \delta(\omega_0 - \omega_{Weyl})$. This is exactly the result of the manuscript under consideration.

In order to elucidate this point better I have plotted the cross section spectra for different detunings $\omega_0 - \omega_{Weyl}$, see the attached pdf. It is clear from the calculation that (i) the width of the peak decreases and (ii) the height increases at the Weyl point which reflects decrease of the spontaneous decay rate $G \Gamma$. The area under the peak stays the same. I believe that the parameters in my calculation illustrate the points (i) slightly better than in Fig. 4a of the manuscript, where the change of the linewidth with the detuning is not resolved.

To summarize, I recommend the Authors to present and discuss carefully and explicitly frequency dependences of *both* spontaneous spectral linewidth and integral cross section (not only the cross section) as functions of $\omega_0 - \omega_{Weyl}$ and $\omega_q - \omega_0$, both in the main text and in the Supplementary. No big changes are required in the structure of the manuscript. However, such discussion would be really helpful for future readers and would justify the concluding statement “Here, we explicitly show the general conservation law of cross section and its connection to the dispersion relation”.

Response: We thank the reviewer for clarifying these very helpful suggestions that were a little vague to us in the first report. Now we fully appreciate it. We have incorporated a few paragraphs in both main manuscript and supplementary on the spectral features of the spontaneous emission rate and the integral cross section. Indeed, we found these discussions strengthen and broaden the conclusion. We’d like to acknowledge the reviewer for this helpful suggestion in our acknowledgement session.

Specifically, here we added the following discussion in the main manuscript :

“

Figure 4. Resonant scattering cross section in Weyl photonic crystal. (a) Diverging resonant scattering cross section is realized around the Weyl frequency. Isosurfaces have an ellipsoidal shape (insets) with its color indicating the value of the cross section; shrinking isosurface leads to increasing cross sections around the Weyl frequency. The results from quantum scattering simulation (red circles) agree well with the prediction based on the band structure (blue dashed line). The cross section is normalized by the average cross section in free space $\bar{\sigma}_0 = \lambda^2/\pi$. It scales as $\sigma \sim 1/\Delta\omega^2$. (b) Spectrum of average scattering cross section for TLSs with different transition frequencies. For simplicity, the spontaneous emission rates of the TLSs in vacuum are chosen to be $10^{-4}\omega_0$. We also assume the dipole moments and locations of the TLSs are the same.

As the TLS's transition frequency approaches the Weyl point, *i.e.*, $\omega_0 \rightarrow \omega_{Weyl}$, the isosurface shrinks in size, as illustrated by the insets of Fig. 4a. The conservation law leads to an increasing $\sigma(\mathbf{q})$, as shown by stronger colors. Near the Weyl frequency, the average cross sections $\bar{\sigma}$ is enhanced by three orders of magnitude compared to that in free space, eventually diverging at the Weyl point (Fig. 4a). The analytical prediction from Eq. 2 and the area of the isosurface agree very well with predictions from numerical simulation (circles in Fig. 4a).

Weyl point greatly enhances the cross section at the resonant frequency. However, it comes at the price of suppressed cross section away from the resonant frequency. Next, we discuss the spectral feature of the average cross section $\bar{\sigma}(\omega)$ for a given TLS. The spectral dependence is shown in the Supplementary Information as

$$\bar{\sigma}(\omega) \sim \frac{(\omega_0 - \omega_{Weyl})^2 \wp}{(\omega_0 - \omega)^2 + \wp^2 \frac{(\omega_0 - \omega_{Weyl})^4}{4}} \quad (4)$$

Here \wp is a constant that depends on the local electric field at the position of the TLS, but does not vary significantly with frequency. In order to derive Eq. 4, we use the fact that the spontaneous decay rate is proportional to $(\omega_0 - \omega_{Weyl})^2$ (See Eq. S32 in SI). At the resonance when $\omega_0 - \omega = 0$, the average cross section scales as $1/(\omega_0 - \omega_{Weyl})^2$, which increases as the resonant frequency moves closer to the Weyl point. However, away from the resonance when $|\omega_0 - \omega| \gg |\omega_0 - \omega_{Weyl}|$, Eq. 4 reduces to $\bar{\sigma}(\omega) \sim (\omega_0 - \omega_{Weyl})^2 / (\omega_0 - \omega)^2$, which shows that being close to the Weyl point suppresses the cross section. In Fig. 4b, we calculate the spectra for 3 different TLSs with their transition frequencies approaching the Weyl point (black dashed line). While the peak value of the cross section grows, the full width at half maximum (FWHM) of the spectrum decreases. The spectral integration of the cross section remains around a constant (see derivation in the Supplementary Information).

”

We have also added a new section in the Supplementary Information to explicitly discuss the spectral linewidth. It reads as:

“

3. Frequency dependence of the spectra of the average cross section

In the main text, we focus on the maximum of the average cross section when $\omega_q = \omega_0$ and briefly discuss the spectrum of the average cross section in Fig. 4. Here we will discuss more details about the spectrum of the average cross section.

The average cross section can be directly calculated by using Eq. (S30). To simplify the discussion, here we will assume that the isosurface around each Weyl point is identical and isotropic, *i.e.*, $|q_{m,n}| = |\omega_{q_{m,n}} - \omega_{Weyl}|/v$. Recall that $(g_0)^2 = d^2 \omega_0 / 2\hbar \epsilon_0 L^3$ and d is the dipole moment of the TLS. Thereby, the spontaneous emission rate Γ can also be calculated as

$$\begin{aligned}
\Gamma &= 2 \sum_{m=1}^M \sum_{n=1}^N \frac{(g_{m,n})^2 L}{v} = \frac{(g_0)^2 L^3 (\omega_0 - \omega_{Weyl})^2}{4\pi^2 v^3} \sum_{m=1}^M \oint \cos \theta_{m,n} f_{m,n} d\Omega_{m,n} \\
&\quad \frac{d^2 \omega_0 (\omega_0 - \omega_{Weyl})^2}{8\hbar \epsilon_0 \pi^2 v^3} \sum_{m=1}^M \oint \cos \theta_{m,n} f_{m,n} d\Omega_{m,n} \\
&= \mathcal{P} (\omega_0 - \omega_{Weyl})^2 \#(S32)
\end{aligned}$$

where $\Omega_{m,n}$ is the solid angle associated with $q_{m,n}$. For simplification, we define a coefficient $\mathcal{P} = d^2 \omega_0 \sum_{m=1}^M \oint \cos \theta_{m,n} f_{m,n} d\Omega_{m,n} / 8\hbar \epsilon_0 \pi^2 v^3$. Around the Weyl point, the coefficient \mathcal{P} does not vary significantly with frequency. As a result, the spontaneous emission rate Γ scales as $(\omega_0 - \omega_{Weyl})^2$.

Next, we integrate the cross section on the isosurfaces and then divide the integration by the area of the isosurfaces. The spectra of the average cross section then is given by

$$\begin{aligned}
\bar{\sigma}(\omega_q) &= \frac{(g_0)^2 L^3 \sum_{m=1}^M \oint \cos \theta_{m,n} f_{m,n} d\Omega_{m,n}}{v \sum_{m=1}^M \oint d\Omega_{m,n}} \frac{\Gamma}{(\omega_0 - \omega_q)^2 + \frac{\Gamma^2}{4}} \\
&= \frac{d^2 \omega_0}{8M\hbar \epsilon_0 \pi v} \sum_{m=1}^M \oint \cos \theta_{m,n} f_{m,n} d\Omega_{m,n} \frac{\Gamma}{(\omega_0 - \omega_q)^2 + \frac{\Gamma^2}{4}} \\
&= \frac{\pi \mathcal{P}^2 v^2}{M} \frac{(\omega_0 - \omega_{Weyl})^2}{(\omega_0 - \omega_q)^2 + \frac{\mathcal{P}^2 (\omega_0 - \omega_{Weyl})^2}{4}} \#(S33)
\end{aligned}$$

Note the prefactor $\pi \mathcal{P}^2 v^2 / M$ does not vary significantly around the Weyl point. On resonance, *i.e.*, $\omega_q = \omega_0$, the resonant average cross section scales as $1/(\omega_0 - \omega_{Weyl})^2$ and increases drastically as $\omega_0 \rightarrow \omega_{Weyl}$. In great contrast, the cross section of off-resonance frequencies scales as $(\omega_0 - \omega_{Weyl})^2 / (\omega_0 - \omega_q)^2$ and is great suppressed around the Weyl point.

Meanwhile, it's clear to see that the integration of the average cross section $\int_{-\infty}^{\infty} \bar{\sigma}(\omega_q) d\omega_q$ is roughly a constant, which can be shown as

$$\begin{aligned} \int_{-\infty}^{\infty} \bar{\sigma}(\omega_q) d\omega_q &= \frac{p^2}{M\pi v^2} \frac{(\omega_0 - \omega_{Weyl})^2}{(\omega_q - \omega_0)^2 + \frac{p^2(\omega_0 - \omega_{Weyl})^4}{4}} \\ &= \frac{2p\pi^2 v^2}{M} \#(S34) \end{aligned}$$

where $2p\pi^2 v^2/M$ remains around the same around the Weyl point.
”

REVIEWERS' COMMENTS:

Reviewer #2 (Remarks to the Author):

My comments have been properly addressed and I do recommend the revised manuscript for publication in the current form